# Quality Aspects in the Compounding of Plastic Recyclate

**Maximilian Auer** [1,*] ⃝, **Jannick Schmidt** [1], **Jan Diemert** [2], **Gabriel Gerhardt** [1], **Maximilian Renz** [1], **Viola Galler** [1] **and Jörg Woidasky** [1] ⃝

1 Institut für Industrial Ecology, Pforzheim University of Applied Science, Tiefenbronner Straße 65, 75175 Pforzheim, Germany

2 Department of Polymer Engineering, Fraunhofer Institute for Chemical Technology, Joseph-von-Fraunhofer-Straße 7, 76327 Pfinztal, Germany

* Correspondence: maximilian.auer@hs-pforzheim.de

**Abstract:** Compounding is the final processing step for quality adjustment and control before recycled thermoplastic polymer material can be introduced into production processes. Motivated by the need for higher recyclate shares, the research question is which quality problems recycling compounders are encountered in practice, where they occur, and which mitigation options might be reasonable. Therefore, an online survey with 20 recycling compounders based in Germany was conducted asking about typical processing steps and processed materials, test procedures for quality assurance, quality problems, and possibilities for reducing quality problems. Results show that compounders mainly name impurities and contaminations of the input material as challenging and the reason for quality problems. The study shows that the problems are not dependent on the material input type. Quality problems occur along the entire secondary value chain, with companies manufacturing components themselves being particularly affected. The composition determination of the input materials helps to minimize quality problems.

**Keywords:** plastic; recycling; compounding; circular economy; quality assurance; polymer testing; thermoplastic material; material properties

## 1. Introduction

The use of recyclates in the production of new plastic goods is an important contribution to a circular economy, as it enables closing material cycles and reduction of dependencies on fossil raw materials and greenhouse gas emission minimization [1]. Accordingly, strengthening recyclate quotas is on the political agenda of both the European Union and Germany. The European Commission and the Circular Plastics Alliance have set themselves the goal to reuse 10 million tons of recycled material in plastic goods across Europe by 2025 [2]. With the *Circular Economy Action Plan*, the European Commission has demanded that, until 2030, all plastic packaging become recyclable or reusable and announced that it will propose mandatory content requirements for the use of recyclate in key plastic products such as packaging, building materials, and vehicles until 2021/2022 [3]. Germany is also aiming to strengthen the use of recyclate by demanding a mandatory plastics recycling rate of 63% by the end of 2022 [4] and with the *5-Point Plan for Less Plastic and More Recycling* from November 2019, initiated by the Federal Ministry for the Environment, Nature Conservation and Nuclear Safety (BMU) [5].

In 2021, 6.31 million t of plastic waste in the form of post-consumer (5.44 million t) and post-industrial (0.87 million t) waste was collected in Germany, resulting in 2.29 million t of recyclate (post-consumer: 1.27 million t; post-industrial: 0.38 million t; byproducts 0.64 million t) after mechanical recycling and subtraction by exports and process losses [6]. At the same time, the German plastic processing industry used 14 million t of polymers, resulting in a recyclate rate of 16.3% across all industries (e.g., construction, packaging,

vehicles) [6]. Compared to the recyclate rate of 13.7% in 2019, this is an increase of 2.6 percentage points [6].

However, the recyclate share in new products is still low, which has its reasons. First and foremost, despite all the political interest, there are still no legally prescribed quotas for recyclates in the production of plastic goods, as this is a significant intervention in the market. This intervention is potentially associated with risks for product quality and safety as well as for raw materials and product prices. Otherwise, recyclate rates are unlikely to increase, as there is little incentive for investment and research with regulatory requirements lacking. In addition, manufacturers often continue to prefer to use primary materials, as recyclates are often more expensive and are not available in sufficient quantity and quality [7]. This is ultimately due to technical obstacles in reprocessing (sorting, handling, etc.) and because the focus is often set on the optimization of individual processing steps. The lack of quality affects the properties of end products negatively, which causes recyclates not to be returned into original products but to be downcycled into secondary products, such as building materials [1]. Furthermore, legal constraints also prevent the use of recyclates. For example, the use of recyclates for the production of food packaging, with regard to the decontamination of remaining contaminants that could endanger human health, is highly regulated in the European Union [6].

For post-consumer plastic recycling, the main steps are collection and transport, sorting, and a subsequent recycling step [8]. This recycling step in most cases covers the optical and physical separation of polymers, but also rinsing processes, and it yields a thermoplastic recyclate after regrinding or compounding [9]. Thus, the recycling compounding process is the final process for quality adjustment and control before the waste material can be introduced into production processes, which makes it the focus of this research work.

In this context, and motivated by the need for a higher recyclate rate, the research question for this work is which quality problems recycling compounders encounter in practice and where they occur. For this purpose, a study was carried out with recycling compounders based in Germany. They were asked about the typical processing steps and processed materials, the test procedures for quality assurance, the quality problems that occur, and possibilities for reducing these quality problems. This provides an up-to-date picture of problems that occur in the final stage of plastic processing. The results from the survey are supplemented and validated with data from the literature.

## 2. Current Problems in the Recycling Process

In Germany, mechanical recycling is used for both the reprocessing of post-consumer and post-industrial plastic waste [10]. In 2019, 46.4% (2.92 million t) of German post-consumer and post-industrial waste was mechanically processed [11]. The widespread use of mechanical treatment of plastic waste is due to its effectiveness in terms of time, economic cost, carbon footprint, and environmental impact [1]. Especially in comparison to chemical recycling, mechanical recycling emits fewer greenhouse gases [12]. The processing steps for mechanical solid plastic waste treatment are essentially separation and sorting by shape, density, size, color, or chemical composition; baling; washing to remove impurities; shredding; and finally compounding and pelletizing [13]. With proper control of processing conditions, many polymers can undergo multiple cycles of mechanical recycling without a significant loss of performance [1]. However, the limitations of mechanical recycling arise, as individual problems and challenges can occur in each processing step. Generally, impurities and contaminations, non-miscibility of polymer shares, and degradation lead to loss of polymeric recyclate properties and result in deficient end-product properties, such as surface deflections, heat streaks, poor geometrical accuracy, or discoloration [1,13,14]. Table 1 summarizes typical recycling problems and their causes.

**Table 1.** Current challenges in the plastics recycling process and their causes.

| Cause | Current Problems in the Recycling Process | Source |
|---|---|---|
| Identification problems due to near-infrared (NIR) spectrometry | Faulty NIR detection due to lightweight packaging being wet or lying in/over each other | [13–16] |
| | Black plastics | [13,14,16–18] |
| | Labeling (>30% of total packaging area) prevents NIR detection | [13–16] |
| | Multilayer packaging (side facing the sensor is detected) | [13,19] |
| Handling problem | Cylindrical (rolling) geometries | [14] |
| | Flexible films with a geometry smaller than 297 mm × 420 mm (Din A3; low density and low bulk density) | [19,20] |
| | Incorrect discharging | [20] |
| | Sorting of small geometries smaller than 20 mm × 20 mm | [14] |
| Density ranges prevent float-sink separation | Separation of foamed/non-foamed elastomer components | [21] |
| | Non-polyolefins with a density smaller than 1 g/cm$^3$ | [14,21,22] |
| | Elastomer components with a density greater than 1 g/cm$^3$ | [21] |
| | Polyolefins with a density greater than 0.995 g/cm$^3$ (filled polymers) | [21,23] |
| Non-miscibility of different polymers | Cross-linked polyethylene (PE-X) components | [21] |
| | Sorting of expanded polystyrene (EPS) | [21] |
| | Polyethylene terephthalate amorphous (PET-A) copolymers | [21] |
| | Identification of plastic types (e.g., polyethylene terephthalate amorphous/glycol-modified (PET-A/G); polypropylene homopolymer/copolymer (PP-H/C)) | [21] |
| | Incompatible components (PET-G; Polyoxymethylene (POM)) | [21] |
| | Polyvinyl Chloride (PVC)/PET-G labels or sleeves | [15,21,24] |
| | Use of barrier coatings (polyamide (PA), polyvinylidene chloride (PVDC), metallization, silicon oxide) in multilayer packaging | [14,15,21] |
| | Separation of PET and PET-PE (multilayer) trays | [25] |
| Impurities | Coatings and varnishes | [13] |
| | Direct printing (extensive; more than "best before end") | [22,26] |
| | Silicone components, not separable | [21,22] |
| | Non-water-soluble adhesives in conjunction with wet-strength labels or radio-frequency identification (RFID) tags | [18,21,23] |
| | Plastic-coated fiber materials | [15] |
| Quality problems on the recyclate | Recyclability of polymers limited by mechanical recycling and degradation | [27] |
| | Discoloration | [1,14] |
| | Odor formation | [1,28] |
| | Undesirable fate of additives (color pigments, fillers, stabilizers, etc.) in the recyclate | [1] |
| Other | Lack of data about the lightweight packaging waste stream | [29] |
| | Differentiation of packaging content: food/nonfood; personal care; cleaning agents; other | [15,30] |
| | Transfer of responsibility to consumers for separation of packaging components consisting of different materials (e.g., aluminum lid and paper band of a polystyrene (PS) cup) | [14] |
| | Packaging with large amounts of residual fill (design prevents residual emptying) | [14,26] |

## 2.1. Challenges in the Identification and Sorting of Plastic Waste

For the treatment of mixed plastic waste, sorting is a crucial processing step. The resulting grade of purity has a major influence on the downstream process of compounding the plastic waste and affects the recyclate quality [13].

### 2.1.1. Problems in Near-Infrared Identification

The sorting of lightweight packaging is essentially based on the identification of relevant inherent packaging properties such as product geometry, conductivity, or absorption properties [31]. Relevant sub-processes of near-infrared (NIR) sorting comprise signal

generation (reflection), signal measurement and assessment (signal interpretation), and plastic product handling for separation. NIR sorting technology is most commonly used for this purpose [32]. The sorting performance of NIR sorters is limited by high belt speeds and the associated short measuring times, as well as by overlapping, black, dirty, wet, or fully printed packaging surfaces, and by labels that cover more than 30% of the packaging and where the polymers differ [13,14]. Further sorting tasks where NIR sorters are reaching their limit or are not capable are the differentiation of food and non-food packaging [33], the sorting of brand-specific materials for the implementation of extended producer responsibility [34], or the identification of certain plastic types of the most important polymers, such as polyethylene terephthalate amorphous/glycol-modified (PET-A/G) and polypropylene homopolymer/copolymer (PP-H/C) [21] or certain multilayer packaging [35].

### 2.1.2. Handling

Once the polymers have been correctly identified by NIR, the next step is their handling, mostly by using compressed air nozzles. The handling of flexible films leads to problems here, as they cannot be discharged from the packaging stream with pinpoint accuracy due to their lightweight nature and the unpredictable trajectory that this entails [13,20]. The packaging geometry can also lead to problems. Cylindrical objects in particular roll on the conveyor belts in the opposite direction to the conveying direction and therefore cannot be discharged accurately [14].

### 2.1.3. Float–Sink Separation

Often, the sorted material in separate recycling facilities is first shredded and then sorted by float–sink separation based on their specific density [13]. However, the plastics' density might be modified, for example by adding fillers or by expanding (foamed polymers), which leads to a deviation from the typical density properties [21]. Density changes are also induced using multilayer composite materials (multilayers) [21]. Sorting into monomaterials is then no longer possible due to insufficient density differences or overlapping density ranges [23,36]. Another problem is the presence of other polymers such as elastomers with similar densities, which are then missorted as well [21,22].

### 2.2. Challenges in the Processing of Plastic Waste

After the plastic waste is sorted, it is compounded into recyclates. Any impurity and contamination will lead to potential problems in the compounding process and finally affect the recyclate quality [13]. Essentially, the problems in reprocessing can be divided into the non-miscibility of plastics, impurities in the polymer stream, and the basic degradation of polymer properties.

### 2.2.1. Immiscibility of Different Polymers

Insufficient sorting purity of the fractions due to missorting leads to problems in the compounding process [1,37]. When recycling mixed thermoplastic fractions, different melting points and processing temperatures (Figure 1) lead to quality losses in the recyclate. Processing two melting components with similar processing temperatures may alter the final product properties, such as color [23]. If higher melting components are present they can be separated and removed by a melt filter, but, at the same time, the melt filter is clogged which requires greater effort for cleaning [8,23]. Components with a low melting point in the recyclate will overheat and degrade, which affects the final product's properties, like reduced mechanical and optical properties [13,23,38]. Particularly often, this problem affects the polymers polyethylene (PE), polypropylene (PP), polystyrene (PS), polyamide (PA), and polyethylene terephthalate (PET), which are frequently used for food packaging, as these are not compatible at the molecular level [39]. Compatibility problems in processing can also occur within these polymer groups, for example when the polymer types PET-A and PET-G or PP-H and PP-C are processed together [21].

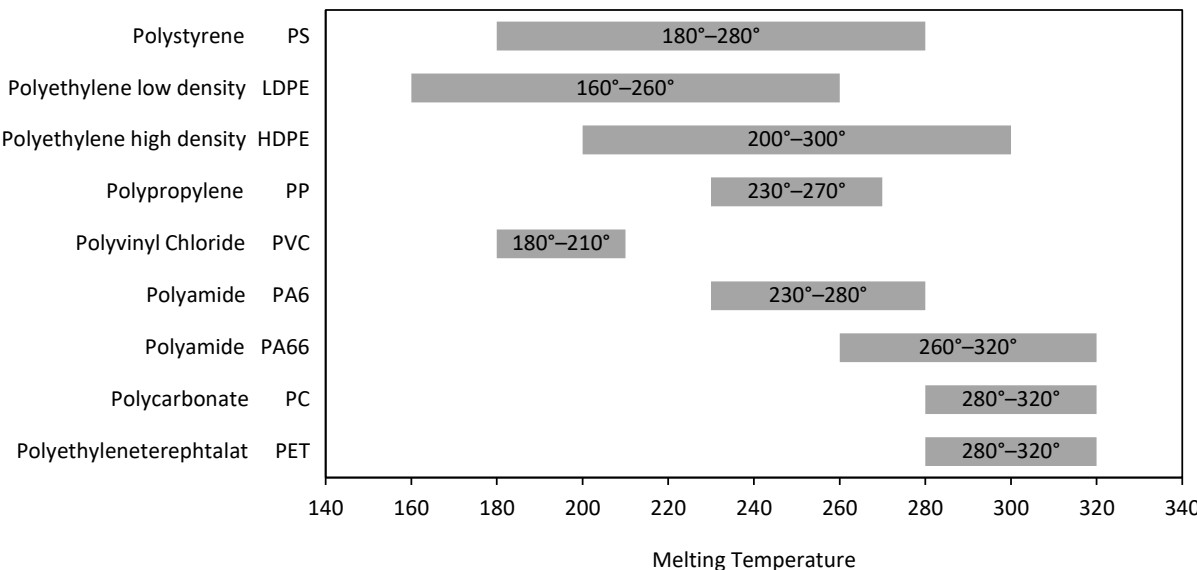

**Figure 1.** Processing temperatures of common thermoplastics [40].

Since contamination by other polymers in household collections is difficult to eliminate, compatibilizing agents are usually used by the processing industry [41]. Compatibilizing agents are added as a third component to the polymer blend to lower the interfacial tension and thereby improve the interfacial adhesion of immiscible polymers [13]. This also leads to an increase in the mechanical properties of the polymer blend [1]. Nevertheless, compatibilizers are also considered as being expensive and having limited efficiency [9].

### 2.2.2. Impurities

Besides immiscible polymers, impurities cause problems in processing as well, as they reduce the recyclate quality, whereby impurities are both non-polymeric materials as well as substances that give specific properties to polymeric packaging, like pigments, coatings, varnishes, printing inks, and additives [13,19,22,42]. These substances are not only induced by the packing material itself, but also by non-separable plastic or paper labels which are wet-strength and therefore both introduce, amongst other things, color pigments into the compounding process [13,21]. Generally speaking, all types of insoluble or non-melting impurities must be removed from the melt as far as possible, as these would inevitably negatively affect the quality and properties of the extrudate [43]. Fractions to be removed here are typically wood, paper, aged rubber particles [44], and wood–plastic composites, which swell when exposed to moisture and thus disrupt the extrusion process due to the resulting water vapor [15]. Especially, lightweight packaging containing aluminum causes major processing problems such as clogging of melt filters and leads to a graying of the recyclate [45].

### 2.2.3. Polymer Degradation Due to Mechanical Processing

The number of reprocessing cycles of mixed plastic waste streams is limited using mechanical recycling [27]. Under the influence of heat, oxidation, light, ionic radiation, hydrolysis, and mechanical shear the polymer chains degrade [46,47]. In particular, heat exposure and mechanical shear during melt processing cause thermal–mechanical degradation of the polymer, which leads to poorer mechanical and rheological properties [48].

### 2.3. External Factors on the Recycling Process

#### 2.3.1. Lack of Data on Plastic Waste

In addition to the challenges already mentioned within the reprocessing of plastic waste, there are also external factors that influence reprocessing. First and foremost, the data basis and the level of detail, particularly in the area of post-consumer waste, are inade-

quate [29]. This is due to the fact that waste volume and composition are subject to constant change due to a variety of framework conditions and factors. For example, socioeconomic parameters (household size, income) or the structure of the disposal area (rural or urban region) influence packaging waste generation [49]. Furthermore, the problem regarding the lack of data on the post-consumer waste stream originates with the packaging manufacturers and distributors. Packaging manufacturers essentially can develop and use any conceivable packaging on the market. On the one hand, this has led to the most efficient and thus cost-optimized packaging solutions possible (e.g., multilayers), and on the other hand to an almost infinite variety of different packaging types consisting of the most diverse materials [35]. The packaging composition is therefore known to the distributor, but this information is not shared along the value chain. This underlines the need for a consistent closed-loop or open-loop approach to plastic packaging recycling between recyclers and manufacturers [50]. However, a corresponding system of closed material cycles can only be created through the exchange of waste-related data and the cooperation of all stakeholders along the supply chain [51]. This is because more in-depth data are key to knowing future forecasts of waste volumes generated and their composition, thus enabling the reprocessing of high-value end products [52], or helping to choose the most appropriate strategy for closing material and product loops [53].

### 2.3.2. Consumers' Understanding of Disposal

Another problem arises when consumers dispose of their plastic waste. A study within Germany showed that 30% of the collected lightweight packaging waste belongs to other recovery routes such as residual waste collection [54]. According to the "German Association for Secondary Raw Materials and Waste Disposal", this is not a problem of willpower, since over 90% of Germans have a positive attitude toward educational campaigns on the subject of waste separation [55], but more a problem of lack of understanding. This is also shown in an online survey conducted by an initiative of the dual systems Germany, where 60% of respondents stated that they lacked detailed knowledge about the correct separation of household waste [56]. In addition, it can be increasingly observed in the packaging market that distributors of packaging are making consumers responsible for separating packaging. For example, in the case of yogurt cups, not only the aluminum lid but also the paper band attached to the cup needs to be removed by the consumer and disposed of accordingly.

Besides the incorrectly disposed-of plastic packaging, there is also the additional problem of food and other residues that remain in the packaging [14]. Both the consumer, who is not emptying the packaging, as well as the packaging design itself, which prevents an effective residual emptying, contribute to this problem, which leads to an unpleasant odor of the recycled material [14,26,28].

### 3. Methods

For a deeper insight into the challenges of polymer compounding, an online survey was conducted in April 2022 with recycling compounders based in Germany. A total of 44 companies were asked to participate in the survey by e-mail. The companies were contacted based on existing contacts as well as on the results of an online search. Of the 44 companies that were contacted, 20 fully completed the questionnaire. The questionnaire comprises four main parts with 21 questions. Questions are mainly multiple-choice with multiple answers, but also single-choice, Likert scale, Boolean, and free text. The four main parts include questions on processed materials and processing steps (A), quality assurance measures (B), quality problems (C), and measures to prevent quality problems (D) which are presented in Table 2.

**Table 2.** Overview of questions from the survey.

| No. | Question | Answer Type |
|---|---|---|
| A | Processed materials and processing steps | |
| A1 | What is the composition of your input material stream? | Multiple choice |
| A2 | In what form do you get your input material stream delivered? | Multiple choice |
| A3 | Which steps of the secondary value chain do you represent? | Multiple choice |
| A4 | What products do you manufacture? | Multiple choice |
| A5 | Which processing equipment do you mainly use in the re-granulation/compounding? | Multiple choice |
| A6 | Does your company process standard thermoplastics? If yes, which ones? | Multiple choice |
| A7 | Does your company process technical thermoplastics? If yes, which ones? | Multiple choice |
| A8 | Does your company process any other plastics? If yes, which ones? | Multiple choice |
| B | Quality Assurance Measures | |
| B1 | What quality assurance do you perform on your input materials? | Multiple choice |
| B2 | What quality assurance do you perform on your output materials? | Multiple choice |
| B3 | Where do you use continuously operating online procedures for quality assurance? | Multiple choice |
| B4 | Which online quality assurance procedures do you use? | Free text |
| C | Quality problems | |
| C1 | How often do quality problems occur? | Likert scale |
| C2 | In my opinion, the quality problems that are relevant for me mainly appear at? | Multiple choice |
| C3 | Please name your top two quality problems. | Free text |
| C4 | How are quality problems usually detected? | Multiple choice |
| D | Measures to prevent quality problems | |
| D1 | Could these quality problems be avoided by better characterization of the input material streams? | Boolean |
| D2 | What makes a better characterization? | Free text |
| D3 | Could these quality problems be avoided by better monitoring of the granules produced? | Boolean |
| D4 | How can better monitoring be achieved? | Free text |
| D5 | Do you plan further investments in the area of quality assurance? | Multiple choice |

Figure 2 shows the allocation of the questions along the process chain. The survey aimed to identify typical problems in practice, where they occur, and whether there are measures that help to solve these problems.

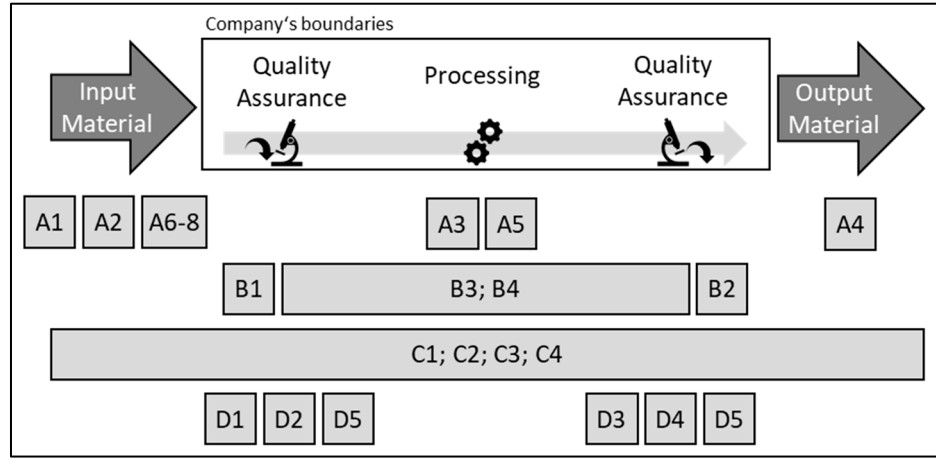

**Figure 2.** Allocation of the questions along the process chain.

For data evaluation, statistical correlations between certain aspects of the study were examined. To examine the dependence of two variables, the Fisher's exact test or linear regression analysis with the coefficient of determination $R^2$ was applied, and the t-test was used to compare two arithmetical mean (M) values. Additionally, the standard deviation (SD) is provided. Before performing the t-test, Levene's test was carried out to assess the equality of variance. The null hypothesis was rejected if the *p*-value (probability) was greater than the significance level of $\alpha = 0.1$. The quantitative evaluations are supplemented

with qualitative results from the study. For improved readability, the questions discussed are abbreviated (e.g., A2 × C1).

## 4. Results

### 4.1. Processed Materials and Processing Steps

The input material (A1) of the participating companies is composed of post-industrial waste (42%), post-consumer waste (16%), or both (42%) types of waste Table 3. Most of the companies' input materials (A2) are already sorted (63%—12 in total). A total of 47% of the companies exclusively process sorted input materials, whereas 11% process exclusively unsorted materials.

**Table 3.** Material input for recycler and compounder.

| Material Input | Post-Consumer | Post-Industrial | Both | Total |
|---|---|---|---|---|
| SORTED | 11% | 32% | 5% | 47% |
| UNSORTED | 5% | 0% | 5% | 11% |
| REGRANULATE | 0% | 5% | 0% | 5% |
| SORTED AND REGRANULATE | 0% | 0% | 11% | 11% |
| SORTED AND UNSORTED | 0% | 5% | 11% | 16% |
| SORTED, UNSORTED AND REGRANULATE | 0% | 0% | 11% | 11% |
| TOTAL | 16% | 42% | 42% | 100% |

A combination of both questions (A1; A2) shows that companies that process post-industrial waste use in particular sorted input materials (six out of eight companies). On the other hand, unsorted material is evenly distributed over post-consumer and post-industrial waste.

Asking about the processing steps of the secondary value chain a company covers (A3), only 60% of the companies answered that they carry out further sorting steps in-house. Milling (90%), granulation (85%), reformulation (75%), and granulate blending (75%) are typical processing steps. Pre-drying takes place at 25% of the companies and component manufacturing at 15%. The number of processing steps within the companies ranges from one to six steps with an arithmetical mean of 4.3 (median: 4.5). It is noticeable that the process step of sorting is not exclusively carried out for unsorted input materials, but also for sorted ones.

The produced goods (A4; output material) correspond to the processing steps conducted by the companies. In total, 15% manufacture products, for example by injection molding, 65% produce re-granulate, and 80% produce new compounds with their own formulations. One company (5%) answered that their materials output is ground material.

Regarding the question of which processing equipment is mainly used (A5, multiple choice), 75% answered that twin-screw extruders and 35% that single-screw extruders are used. Furthermore, 65% use underwater granulation, 55% strand granulation and 15% eccentric pelletizer. Melt degassing is used by 90% of the companies surveyed, and a melt filter by 65%.

Figure 3 shows the polymers processed by the companies (A6–A8). The first columns show the absolute number of companies and their percentage processing the polymer groups with the corresponding polymer types. It can be seen that both standard thermoplastics (85%) and technical thermoplastics (75%) are being processed often, whereas only 20% responded that they are processing other polymers such as bio-based or bio-degradable polymers.

| | | Number of companies processing this polymer | Percentage | Standard Thermoplastic | PP | HDPE | LDPE | LLDPE | ABS | PS | HIPS | PVC | PET | Other standard thermoplastic | Technical Thermoplastic | PA6 | PA66 | POM | PMMA | PC/ABS | PC | Other Technical Thermoplastic | Other Polymers | Bio-based polymer | Bio-degradable polymer | Other Polymers |
|---|---|---|---|---|---|---|---|---|---|---|---|---|---|---|---|---|---|---|---|---|---|---|---|---|---|---|
| Standard Thermoplastic | | 17 | 85% | 100% | 82% | 59% | 18% | 18% | 47% | 35% | 12% | 0% | 35% | 12% | 71% | 53% | 53% | 24% | 18% | 47% | 47% | 35% | 24% | 18% | 6% | 6% |
| Polypropylene | PP | 14 | 70% | 100% | 100% | 57% | 14% | 14% | 57% | 43% | 14% | 0% | 36% | 14% | 79% | 64% | 64% | 21% | 21% | 57% | 57% | 43% | 21% | 21% | 7% | 0% |
| High-Density Polyethylene | HDPE | 10 | 50% | 100% | 80% | 100% | 20% | 20% | 60% | 40% | 10% | 0% | 40% | 10% | 80% | 50% | 50% | 20% | 0% | 40% | 40% | 30% | 30% | 20% | 0% | 10% |
| Low-Density Polyethylene | LDPE | 3 | 15% | 100% | 67% | 67% | 100% | 100% | 33% | 33% | 33% | 0% | 67% | 33% | 67% | 33% | 33% | 0% | 0% | 33% | 33% | 0% | 33% | 33% | 0% | 0% |
| Linear Low-Density Polyethylene | LLDPE | 3 | 15% | 100% | 67% | 67% | 100% | 100% | 33% | 33% | 33% | 0% | 67% | 33% | 67% | 33% | 33% | 0% | 0% | 33% | 33% | 0% | 33% | 33% | 0% | 0% |
| Acrylnitril-Butadien-Styrol | ABS | 8 | 40% | 100% | 100% | 75% | 13% | 13% | 100% | 75% | 25% | 0% | 50% | 13% | 100% | 88% | 88% | 25% | 13% | 63% | 63% | 75% | 25% | 25% | 0% | 0% |
| Polystyrene | PS | 6 | 30% | 100% | 100% | 67% | 17% | 17% | 100% | 100% | 33% | 0% | 67% | 17% | 100% | 83% | 83% | 17% | 17% | 50% | 50% | 50% | 17% | 17% | 0% | 0% |
| High Impact Polystyrene | HIPS | 2 | 10% | 100% | 100% | 50% | 50% | 50% | 100% | 100% | 100% | 0% | 100% | 50% | 100% | 100% | 100% | 50% | 50% | 50% | 50% | 50% | 50% | 50% | 0% | 0% |
| Polyvinyl Chloride | PVC | 0 | 0% | 0% | 0% | 0% | 0% | 0% | 0% | 0% | 0% | 0% | 0% | 0% | 0% | 0% | 0% | 0% | 0% | 0% | 0% | 0% | 0% | 0% | 0% | 0% |
| Polyethylene Terephthalate | PET | 6 | 30% | 100% | 83% | 67% | 33% | 33% | 67% | 67% | 33% | 0% | 100% | 17% | 83% | 67% | 67% | 17% | 17% | 50% | 50% | 50% | 17% | 17% | 0% | 0% |
| Other standard thermoplastic | | 2 | 10% | 100% | 100% | 50% | 50% | 50% | 50% | 50% | 50% | 0% | 50% | 100% | 50% | 50% | 50% | 0% | 0% | 0% | 0% | 0% | 50% | 50% | 0% | 0% |
| Technical Thermoplastic | | 15 | 75% | 80% | 73% | 53% | 13% | 13% | 53% | 40% | 13% | 0% | 33% | 7% | 100% | 67% | 67% | 27% | 33% | 60% | 73% | 40% | 27% | 27% | 7% | 7% |
| Polyamide 6 | PA6 | 10 | 50% | 90% | 90% | 50% | 10% | 10% | 70% | 50% | 20% | 0% | 40% | 10% | 100% | 100% | 100% | 30% | 30% | 80% | 80% | 60% | 30% | 30% | 10% | 0% |
| Polyamide 66 | PA66 | 10 | 50% | 90% | 90% | 50% | 10% | 10% | 70% | 50% | 20% | 0% | 40% | 10% | 100% | 100% | 100% | 30% | 30% | 80% | 80% | 60% | 30% | 30% | 10% | 0% |
| Polyoxymethylene | POM | 4 | 20% | 100% | 75% | 50% | 0% | 0% | 50% | 25% | 25% | 0% | 25% | 0% | 100% | 75% | 75% | 100% | 50% | 75% | 75% | 100% | 75% | 50% | 25% | 25% |
| Poly(Methyl Methacrylate) | PMMA | 5 | 25% | 60% | 60% | 0% | 0% | 0% | 20% | 20% | 20% | 0% | 20% | 0% | 100% | 60% | 60% | 40% | 100% | 60% | 60% | 40% | 20% | 20% | 20% | 0% |
| Polycarbonate/Acrylnitril-Butadien-Styrol | PC/ABS | 9 | 45% | 89% | 89% | 44% | 11% | 11% | 56% | 33% | 11% | 0% | 33% | 0% | 100% | 89% | 89% | 33% | 33% | 100% | 100% | 67% | 22% | 22% | 11% | 0% |
| Polycarbonate | PC | 11 | 55% | 73% | 73% | 36% | 9% | 9% | 45% | 27% | 9% | 0% | 27% | 0% | 100% | 73% | 73% | 27% | 45% | 82% | 100% | 55% | 18% | 18% | 9% | 0% |
| Other Technical Thermoplastic | | 7 | 35% | 86% | 86% | 43% | 0% | 0% | 86% | 43% | 14% | 0% | 43% | 0% | 86% | 86% | 86% | 57% | 29% | 86% | 86% | 86% | 43% | 43% | 0% | 0% |
| Other Polymers | | 4 | 20% | 100% | 75% | 75% | 25% | 25% | 50% | 25% | 25% | 0% | 25% | 25% | 100% | 75% | 75% | 75% | 75% | 50% | 50% | 75% | 100% | 75% | 25% | 25% |
| Bio-based polymer | | 3 | 15% | 100% | 100% | 67% | 33% | 33% | 67% | 33% | 33% | 0% | 33% | 33% | 100% | 100% | 100% | 67% | 33% | 67% | 67% | 100% | 100% | 100% | 33% | 0% |
| Bio-degradable polymer | | 1 | 5% | 100% | 100% | 0% | 0% | 0% | 0% | 0% | 0% | 0% | 0% | 0% | 100% | 100% | 100% | 100% | 100% | 100% | 100% | 0% | 100% | 100% | 100% | 0% |
| Others polymers | | 1 | 5% | 100% | 0% | 100% | 0% | 0% | 0% | 0% | 0% | 0% | 0% | 0% | 100% | 0% | 0% | 100% | 0% | 0% | 0% | 0% | 100% | 0% | 0% | 100% |

**Figure 3.** Processed polymers and combinations of polymers being processed together.

The most processed polymers are PP (70%), followed by polycarbonate (PC) (55%) and high-density polyethylene (HDPE), PA6 and PA66 (50% each). Polyvinyl chloride (PVC), on the other hand, is not processed by any of the companies. The companies process between 1 and 13 polymers, with an arithmetical mean of 6.0 (median: 5.5).

Further evaluation can be made of the polymers being processed together. For example, companies that process PP also process HDPE with a frequency of 57% (8 of 14 companies) and companies that process PA6 also process PA66 (100% frequency). It might be expected that companies mainly process polymers that have similar chemical structures, which is the case for PA6 and PA66 or acrylnitril-butadien-styrol (ABS), PS, and high-impact polystyrene (HIPS). Nonetheless, there are also polymers with a low frequency that have similar chemical structures, like low-density polyethylene (LDPE) and HDPE (20% frequency).

### 4.2. Quality Assurance Measures

Nearly all companies surveyed stated that they carry out quality assurance measures in the form of quality checks on the input material (B1); only one company did not provide any information on this (Figure 4). Among the most frequently mentioned test procedures were the melt flow index (15; 75%), ash content determination (12; 60%), density testing (11; 55%), and moisture (10; 50%). The number of input testing procedures used per company ranges from 1 to 24, with an arithmetical mean of 8.9 (median: 7.0).

All the companies also carry out quality controls on the output material (B2). Here, the range of testing procedures used is between 1 and 22, with an arithmetical mean of 8.7 (median: 7.0). In addition to the melt flow index (17; 85%), ash content determination (15; 75%) and tensile testing (15; 75%) are performed most frequently. Other quality control methods used are the Charpy impact test (14; 70%), moisture content test (12; 60%), density test (12; 60%), color measurement (10; 50%), bending test (10; 50%), differential scanning calorimetry (DSC) (9; 45%) and hardness test (9; 45%).

In general, the more test methods a company uses on the input stream, the more test methods are used on the output stream. On input material, test methods to determine the exact polymer are used more frequently than on the output stream. On the other hand, on the output stream, test methods for processability and the material properties which are relevant in the use phase are focused on.

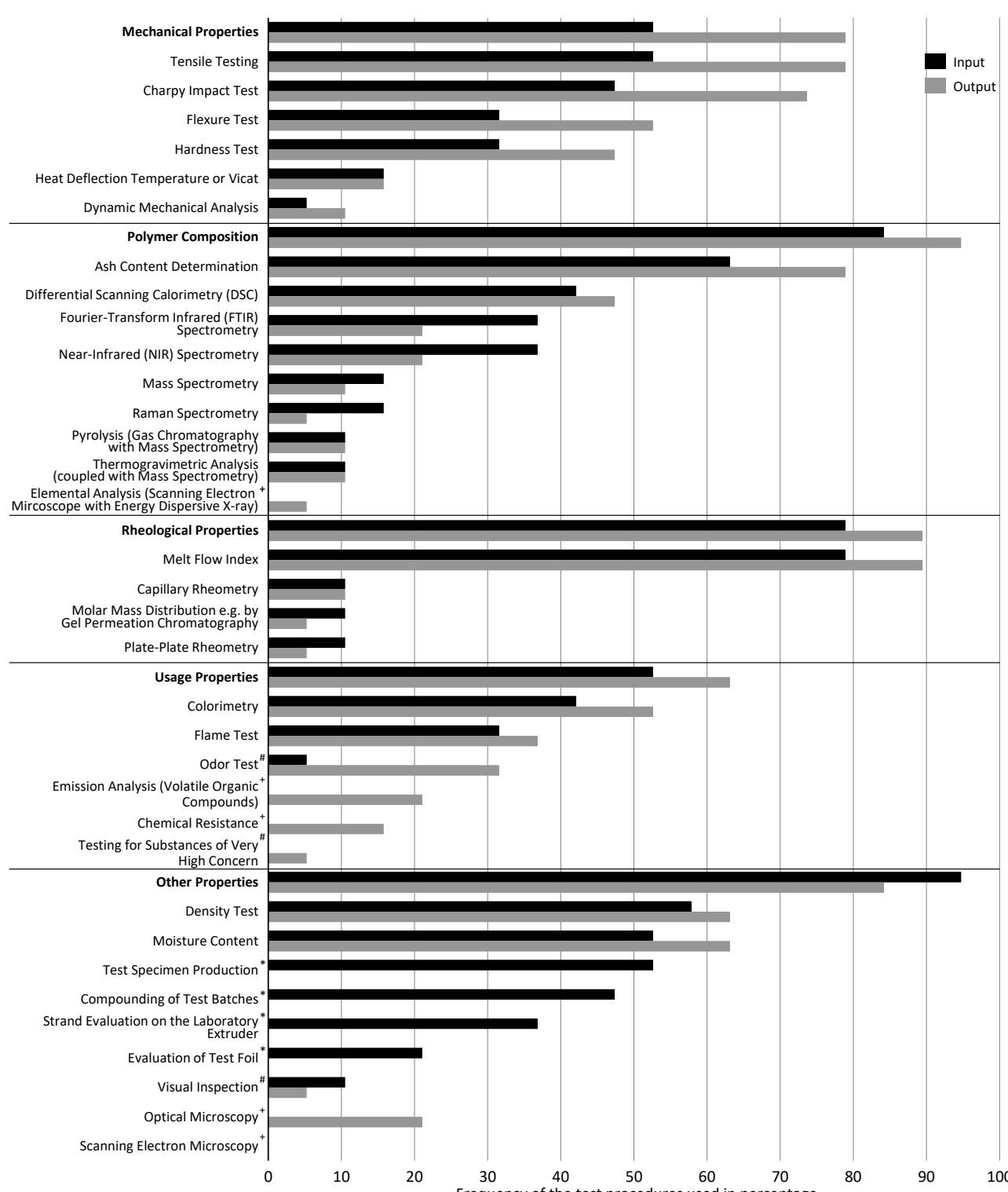

*exlusively reply option for input material +exclusively reply option for output material #free text reply for others

**Figure 4.** Quality assurance measures on input and output materials (B1–B2).

Besides the classical test methods, where a sample is analyzed in the laboratory, online methods (measurement methods directly on the process) are conducted (B3; B4). A total of nine (45%) companies use online methods, whereas one company does not specify where their methods are used. The other eight companies stated that they use online methods in the compounding process (100%), of which five companies (63%) use online

methods exclusively in this processing step. In compounding, online methods are used to monitor and measure pressure, temperature, power consumption, rheology, and color. Furthermore, online methods are used on the finished products (three companies, 38%) and the input material (one company, 13%). Besides the already-mentioned online methods, UV measurements and layer and total thickness measurements are online methods conducted, but they cannot be allocated to a specific processing step.

### 4.3. Quality Problems

The majority of the companies stated that they are very rarely (30%) or rarely (40%) affected by quality problems (total 70%, C1). A total of three companies (total of 15%) were frequently (10%) or very frequently (5%) affected by quality problems. Three companies (15%) responded that no statement is possible.

The evaluation also showed that quality problems occur along the entire process chain (C2). Problems in the view of the respondents are allocated in particular at the beginning of the process chain with waste collection (40%) and sorting (40%). Furthermore, relevant problems occur in washing (15%), shredding (25%), and re-sorting (10%), but also granulation (25%), reformulation (10%), granulate blending (10%), and component production (15%). Only in the drying of the granules no relevant problems arise for any company.

Asked about the most important quality problems (C3), 70% declare that contamination is the main issue. Contamination reaches from incompatible polymers to other materials such as labels, metals, or moisture. Some companies specify that contaminations are caused by fluctuating input purities, which vary by season and sorting plant, and insufficient pre-sorting of the material. Interestingly, all eight companies who stated that quality problems occur in waste collection also responded that contamination is their main problem. Other problems include off-spec output products (25%) where the color mismatches or the geometric accuracy of components is poor, or surface defects, coating formation, or heat streaks occur.

On the question of how quality problems are discovered (C4), one company (5%) stated that they are reported by the supplier. All companies (100%) discover quality problems in-house (through quality assurance measures), and nine companies (47%) state that they are notified by their customers about quality problems.

### 4.4. Preventing Quality Problems

The companies were also asked if quality improvement can be achieved by a better characterization of the input materials (D1; D2) or by better monitoring of the output materials (D3; D4). In total, 45% stated that improved characterization of input materials can reduce quality problems. Clean and mono-material inputs as well as better labeling of the products are mentioned as possible improvements. In addition, better information about the material input may already help to improve the output quality. Furthermore, the introduction of additional quality assurance measures with new testing methods, which are not yet in use, might help to reduce quality problems. However, it was also stated that higher quality cannot be achieved economically due to costs and labor intensity.

On the other hand, only 20% state that the quality problems of the produced output materials could be improved by better monitoring (D3; D4). Here, continuous testing procedures, like online measurements and direct rejection of defective products, might help to reduce quality problems according to the respondents.

Regarding future investments in quality assurance measures (D5), 90% of the companies state that they plan to extend their quality assurance measures either by buying new, not-yet-used, test methods (60%) and/or by replacing existing test methods with new ones (60%). Only 10% stated that they do not plan further investments in quality assurance measures.

### 4.5. Effect on the Appearance of Quality Problems

One question to be answered by this study is which factors influence quality problems in companies and of the finished recyclate. Therefore, it was first examined whether quality problems occur in certain processes of the secondary value chain (A3 × C1). There is no correlation between the increasing number of processing steps a company carries out and the frequency of quality problems arising ($R^2 = 0.038$, $p = 0.0454$). However, it is noticeable that two of the three companies, which frequently complain about quality problems, manufacture components themselves. This correlation is statistically significant ($p = 0.063$), as Fisher's exact test shows. This correlation is also supported by the other questions, whereby these companies state that they have problems with component manufacturing, and that surface defects such as coating formation are among the most significant quality problems (C2; C3). Another quality problem that was frequently mentioned (70%) is the occurrence of contaminants and impurities in the material stream. This suggests that companies that additionally sort the input material are less likely to be affected by quality problems since they remove the problematic substances. Here, however, Fisher's exact test ($p = 0.228$) does not indicate that quality problems occur dependently of sorting.

In this context, one would also assume that the input material itself, i.e., whether post-consumer or post-industrial plastic waste is processed, has an impact on the frequency of quality problems (A1 × C1). In particular, post-consumer plastic waste is considered challenging due to its heterogeneity and impurities. However, no correlation between the input materials and the frequency of quality problems can be observed ($p = 1.000$).

The processing equipment can also affect the quality of recyclates (A5 × C1). For example, the literature indicates that melt filtration can remove solid impurities and melt degassing can remove impurities with low molar mass from the melt in the extrusion process [8]. However, neither melt filtration ($p = 0.515$) nor melt degassing ($p = 1.000$) have a significant influence on the frequency of occurrence of quality problems in the present data set.

Another aspect in which the companies differ greatly is the different test procedures and their number, which are used for quality assurance. It is assumed that a company that uses more test procedures is less likely to be affected by quality problems (B1 × C1; B2 × C1). In principle, however, this correlation between the number of test procedures used on the input or output material flow and the frequency of quality problems cannot be verified. The linear regression analysis for the input test procedures returns a coefficient of determination $R^2 = 0.030$ ($p = 0.509$) and for the output test procedures $R^2 = 0.066$ ($p = 0.318$). The variable frequency of quality problems shows no linear dependency on the variables for numbers of test procedures on the input and the output material.

The test procedures used can be differentiated into five subgroups, as shown in Figure 4. The regression analysis for these subgroups of test procedures on the input material shows that the mechanical ($R^2 = 0.014$, $p = 0.651$), rheological ($R^2 = 0.008$, $p = 0.736$), usage ($R^2 = 0.039$, $p = 0.448$), and other ($R^2 = 0.182$, $p = 0.088$) properties are not linearlu dependent of the frequency of quality problems occurring. For polymer composition, a moderately strong correlation is shown with a coefficient of determination $R^2 = 0.334$ ($p = 0.015$). The fewer test procedures a company uses to determine input material composition, the more frequently it will be affected by quality problems. Since the quality of the input material flow in particular is criticized by the companies, this correlation also seems to be causally consistent. Regarding the testing of the output material, no linear correlation can be found for mechanical ($R^2 = 0.015$, $p = 0.638$), rheological ($R^2 = 0.119$, $p = 0.176$), usage ($R^2 = 0.065$, $p = 0.325$), other properties ($R^2 = 0.111$, $p = 0.190$), or polymer composition ($R^2 = 0.029$, $p = 0.515$).

Furthermore, some factors determine the number of test procedures used. For this purpose, the mean values for the number of test methods used, in each case for the input as well as the output material, were compared with the uses of certain processes and materials.

The input material shows that neither the sorted nor the unsorted input material streams significantly influence the number of test procedures (A2 × B1; A2 × B2). The

picture is different for the origin of the input material stream (A1 × B1; A1 × B2): if post-industrial waste is accepted, the companies use more test procedures for quality assurance than if they do not process post-industrial waste. In the testing of the input stream, the mean values (yes: M = 9.9, SD = 6.99, no: M = 5.3, SD = 4.35) do not differ significantly (t (17) = 1.243; $p$ = 0.231). In contrast, significantly more testing procedures are used at the output stream with an arithmetic mean of 10.3 (SD = 6.02) (t (17) = 1.95, $p$ = 0.068) compared to not processing post-industrial waste (M = 4.3, SD = 2.36). There are a total of four companies that report not processing post-industrial waste. These four companies all indicate that they additionally sort their input material stream.

This indicates that if sorting is carried out as a process step in the secondary value chain within a company, the number of test procedures used is reduced (A3 × B1; A3 × B2). This is also evident in the comparison of the mean values, which shows that companies that sort have an average of 6.8 input test procedures (SD = 5.49) in use, which is less than companies that do not sort (M = 11.8, SD = 7.5), although this difference is not significant (t (17) = −1.66, $p$ = 0.115). However, the difference becomes significant when considering the test procedures on the output material. Companies that additionally sort the input material perform significantly (t (17) = −2.05, $p$ = 0.056) fewer test procedures on the output material (M = 6.6, SD = 4.45) than companies that do not additionally sort their input material stream (M = 11.8, SD = 6.48).

Furthermore, it is noticeable that the difference in the number of test procedures for companies using granulation (input testing: M = 10.0, SD = 6.75; output testing: M = 10.1, SD = 5.84) compared to companies that are not using granulation (input testing: M = 3.0, SD = 0.00; output testing: M = 3.3, SD = 2.52) is significant (input testing: t (17) = 1.75, $p$ = 0.097; output testing: t (17) = 1.94, $p$ = 0.069). The same applies to companies that do reformulation and homogenization (input testing: M = 11.1, SD = 6.37; output testing: M = 11.1, SD = 5.43) compared to companies that do not do reformulation and homogenization (input testing: M = 2.6, SD = 1.67, output testing: M = 3.2, SD = 2.49), where the difference is significant as well (input testing: t (17) = 2.91, $p$ = 0.010, output testing: t (17) = 3.11, $p$ = 0.006). On the other hand, no such significant difference in the number of test procedures can be observed whether companies produce components (input testing: M = 3.7, SD = 2.52; output testing: M = 4.7, SD = 3.22) or not (input testing: M = 9.9, SD = 6.82, t (17) = −1.53, $p$ = 0.145; output testing: M = 9.9, SD = 6.07, t (17) = −1.43, $p$ = 0.172), nor whether a company performs shredding (input testing: M = 8.4, SD = 5.81; output testing: M = 8.8, SD = 5.04) or not (input testing: M = 13.5, SD = 14.85, t (1.036) = −0.49, $p$ = 0.710; output testing: M = 11.5, SD = 14.85, t (1.027) = −0.26, $p$ = 0.838).

In addition, companies that do drying as a part of the secondary value chain use significantly more test procedures on the input material (M = 14.6, SD = 5.55) than companies that do not dry their material (M = 6.9, SD = 5.97, t (17) =2.53, $p$ = 0.022). It is most likely that the polymer processed influences the number of test procedures; e.g., four of the five companies that are drying their input material also process technical thermoplastics, such as PA6/PA66. More test procedures are also used on the output material when a company performs drying (M = 12.4, SD = 5.90) compared to when they do not (M = 7.9, SD = 5.72), even though this difference is not significant (t (17) = 1.51, $p$ = 0.149).

## 5. Discussion

The survey allows some conclusions on the problems that compounders encounter in plastic reprocessing. The assumption that the input material has a significant influence on the frequency of quality problems could not be confirmed by the survey results. Whether a company processes post-consumer or post-industrial input material, or if this material is sorted, unsorted, or already supplied as regranulate, has no significant effect on the frequency of quality problems. This is unexpected, as in different material streams different levels of contamination and foreign material contamination would have been expected, which in turn were named as one of the largest problems by the participating companies.

An analysis of the process steps used in the secondary value chain reveals that component production, in particular, stands out. Companies that are frequently affected by quality problems produce components themselves. This process step apparently is more demanding than, for example, reformulation and homogenization, and quality problems are immediately visible. However, the companies which cover the process step of component production in their company neither use more nor fewer test procedures for quality assurance than companies that reformulate and homogenize granules and compounds.

The additional process step of sorting does not result in a reduction of quality problems. It would have been expected that additional in-house sorting based on its own quality standards should reduce the problems, but, in fact, no difference in the frequency of occurrence of quality problems can be observed. However, the companies trust the in-house sorting more, which is reflected in the number of test procedures used, where companies that sort in-house used fewer test procedures than companies that do not sort in-house.

In general, the number of test procedures used does not seem to allow any conclusions regarding the frequency of occurrence of quality problems. The number of test procedures varied greatly between the companies, but there seems to be a correlation between the number of test procedures on the input and at the output material, where companies that do a lot of testing on the input stream do more testing on the output stream and vice versa. Moreover, the companies are already using the procedures that they consider relevant. This is also confirmed by the question about planned investments, where a large number of companies state that they will only invest in replacing existing equipment rather than buying new, not-yet-used technologies. This can be especially observed in those companies that state that they are frequently or very frequently affected by quality problems—none of these companies plan to invest in new technologies that are not used.

The evaluation of this study shows that testing the polymer composition can reduce the frequency of quality problems. The more test procedures a company uses to determine the material composition of the input material stream, the less frequently it is affected by quality problems. Among the test procedures used for this purpose, the most frequently used are the ash content determination, the DSC measurements, and the NIR measurements. This correlation again matches closely with the perception of the surveyed companies that the input material flow is the most important quality problem.

Furthermore, it is assumed that the use of online procedures can reduce quality problems. Such a correlation could not be discovered in this survey, or even the opposite, since all companies with frequent or very frequent quality problems already used online procedures. In contrast, these companies seem to be much more aware of their quality problems and have already invested in online quality assurance procedures. This may even result in problems becoming known in the first place, which increases the subjective perception of the frequency of the occurrence of quality problems.

Summarizing the statements of the companies, the causes of the quality problems are found in the previous processing steps, such as sorting, as well as collection. When asked about the most frequent quality problems, only a minority stated that these occur as a result of in-house processes which might be eliminated by readjustment. Neither better characterization of input materials nor better monitoring of output material flows can reduce quality problems, according to the majority of companies surveyed.

## 6. Limitations of the Study

The presented survey is subject to limitations. First and foremost, the dataset is relatively small, with 20 respondents only. The data and results presented are valid for the given dataset but not necessarily for the statistical population. Due to the small dataset, the significance level chosen is set at $\alpha = 0.1$, which increases the uncertainty factor. In addition, the answers are subject to the respondents' subjectivity, which led to uncertainty when answering the question about the frequency of quality problems, for example.

## 7. Conclusions

The use of recyclates in the production of new plastic goods is an important contribution to achieving a circular economy, as it enables closing material cycles and reducing dependencies on fossil raw materials and emissions of greenhouse gas [1]. Accordingly, both the European Union and Germany want to strengthen recyclate quotas [3,4]. Nevertheless, currently, only 16.3% of recyclate is used in the plastic processing industries. The reasons for the low recyclate consumption are manifold and range from political interests and legal regulations to economic and qualitative factors.

Currently, the mechanical processing of plastic waste into recyclates is used most frequently in Germany. Polymer reprocessing comprises separation and sorting according to product properties, baling, washing, shredding, compounding, and pelletizing [13]. In the recycling process, a wide variety of problems can occur along the entire value chain, which might affect the final recyclate quality [13]. The problems encountered include but are not limited to the identification problems due to NIR, handling problems, density ranges preventing float–sink separation, non-miscibility of different polymers, and impurities [14].

To provide a more detailed understanding of the challenges in the compounding process of recyclate, a survey was conducted among Germany-based compounding companies. The companies were asked about their processing steps and the materials to be processed, quality assurance measures, quality problems, and preventive measures. The aim was to identify the problems that typically occur and to find out which quality assurance measures can help to reduce these problems.

In conclusion, the study shows that the problems are not dependent on the material input type. Quality problems can occur along the entire secondary value chain, with companies that manufacture components themselves being particularly affected by quality problems. Additional in-house sorting of materials does not lead to a significant reduction in quality problems. The absolute number of test procedures used does not influence the frequency of quality problems either. However, the use of more test procedures to determine the composition of the input stream does contribute to fewer quality problems for a company.

The companies surveyed stated in the study that the input material in particular is to be regarded as the main problem. This includes impurities and contaminations, which negatively influence the output qualities. Improved collection and sorting can help to reduce these problems. However, this does not seem to be economically feasible with current sorting technology and contractual constraints. Looking to the future, and aiming to increase recycling rates and qualities, technological innovation and policy change will be required. If the rules for separate plastics collection for consumers do not change, only disruptive sorting technologies such as image-based methods combined with artificial intelligence can remove impurities, or completely different approaches such as water markers or the use of tracer technologies can sort previously unsortable fractions. Furthermore, politics can either regulate the market and prescribe recycling quotas or promote a higher use of recyclates through financial incentives such as a plastic tax.

**Author Contributions:** Conceptualization, M.A., J.D. and J.W.; methodology, M.A., J.S. and J.D.; validation, J.D. and J.W.; formal analysis, M.A. and V.G.; investigation, G.G. and M.R.; data curation, M.A.; writing—original draft preparation, M.A. and J.S.; writing—review and editing, M.A., J.S., J.D., V.G. and J.W.; visualization, M.A. All authors have read and agreed to the published version of the manuscript.

**Funding:** This research received no external funding.

**Institutional Review Board Statement:** Not applicable.

**Informed Consent Statement:** Not applicable.

**Data Availability Statement:** The data in this paper contains information that allows conclusions about the participating companies and cannot be published for privacy reasons.

**Acknowledgments:** The authors appreciate the support of Andreas Stolzenberg for conducting this survey.

**Conflicts of Interest:** The authors declare no conflict of interest.

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
