# Peer review of "Quality Aspects in the Compounding of Plastic Recyclate"

_recycling, doi:10.3390/recycling8010018_

Round 1

Reviewer 1 Report

The manuscript focus on the main recycling problems that nowadays the compounders encounter, particularly in the final stage of plastic processing. The manuscript considers as case-study companies allocated in Germany and it is well organized, uses statistical analysis and brings new data to this field, however it requires some important corrections before acceptance in this journal, namely:

- In page 1, Abstract section should indicate the number of 20 respondents in this study, in order to indicate the difficulties / limitation of this survey.

- General: The author should carefully provide a version with the clear indication of the figures and the tables to avoid the indication in the text “Error! Reference source not found.”.

- General: Thera are several acronyms that appears in table 1, that are only explain after this table section (RFID, LWP) or are not defined such as NIR or HIPS. The author should carefully revise the manuscript to provide all the acronyms for a correct understanding.

- In page 10, figure 4 the author should correct “NIR spectroskopy”.

- In page 15, Bibliography section. In this study at least 30% of the provided references are from electronic sources in the form of online reports, news, others. Thus, it is recommended that the author double-check the existence of the provide links and indicate a more recent accessed date (for instance Reference 56 accessed on 9 April 2021) to guaranty the access to the potential audience of this study. For instance, the link of Reference 4 is not working to access to the information.

Author Response

Dear Reviewer,

Thank you for taking the time to provide me with feedback on the submitted manuscript. Please see the attached document for the changes made.

Kind regards

Reviewer 2 Report

The paper provides statistical evaluation of the relations between quality of plastic waste and methods of its sorting and problems with quality of recycled materials. The paper is based on the treatment of the literature data and of authors’ own questionary.  According to my feeling, meaning of problems with quality should be specified in more detail. I am not familiar with recycling industry in Germany, but according to my experience, most recycled plastic materials have deficiency in end-used properties which limits their applicability. Therefore, problems with achievement of recycled plastic materials with a higher added value should be at least mentioned.

Some abbreviations in the paper are not explained.

There are some places (marked in the manuscript) where sources of statements should be added.

Author Response

(The authors gave the same response as above.)

Round 2

Reviewer 1 Report

The authors addressed the comments raised by the reviewer´s, however there is a minor revision before can be considered for publication, namely:

It is not recommended in the manuscript to place a table 3 (abbreviations list), placed after the conclusion section. In addition, table 3 appears two times for different tables (page 8 and page 16 respectively). Moreover, is not recommended for instance and after table 1, the author provide the indication to see table 3 without pass the following table 2.

The abbreviation should be providing after the first time that the author refers the nomenclature, and along the manuscript (Differential Scanning Calorimetry (DSC); Energy Dispersive X-ray Analysis (EDX), etc.).

For instance, in the case of the non-mentioned polymers in the text, and for those that are only indicated in Figure 3, the author can provide the other abbreviations in the figure caption.

Author Response

Dear Reviewer,

thank you very much for your suggestions. According to your recommendations, we have introduced the abbreviations the first time they are mentioned in the text. Furthermore, the term including the abbreviation has been added to each table or figure, so that they can now stand alone. Table 4 with the overview of all abbreviations has been removed.

Best regards!